# Genome-Wide Association Analysis Identifies the *PMEL* Gene Affecting Coat Color and Birth Weight in Simmental × Holstein

**DOI:** 10.3390/ani13243821

**Published:** 2023-12-11

**Authors:** Jing Wang, Tingting Fan, Zhenwei Du, Lingyang Xu, Yan Chen, Lupei Zhang, Huijiang Gao, Junya Li, Yi Ma, Xue Gao

**Affiliations:** 1Laboratory of Molecular Biology and Bovine Breeding, Institute of Animal Science, Chinese Academy of Agricultural Sciences, Beijing 100193, China; 82101215390@caas.cn (J.W.); 17862512712@163.com (T.F.); dzw1232023@163.com (Z.D.); xulingyang@caas.cn (L.X.); chenyan0204@163.com (Y.C.); zhanglupei@caas.cn (L.Z.); gaohj111@sina.com (H.G.); lijunya@caas.cn (J.L.); 2Animal Husbandry Institute, Tianjin Academy of Agricultural Sciences, Tianjin 300384, China

**Keywords:** coat color, birth weight, Simmental × Holstein, GWAS, *PMEL*

## Abstract

**Simple Summary:**

Coat color and birth weight are important traits in beef cattle and have a significant impact on breed identification and adult body weight. In this study, genome-wide association analysis (GWAS) of Simmental × Holstein (F1) crossbred cattle revealed that calves with lighter coat color had higher birth weight. Further analysis indicated that mutations in the premelanosome protein (*PMEL*) gene resulted in a lighter coat color, while the birth weights of individuals heterozygous for this gene were significantly higher than those of the pure genotype. Therefore, mutations in *PMEL* can result in a lighter coat color, and lighter coat color may, thus, have a selective effect on calf birth weight.

**Abstract:**

Coat color and birth weight, as easily selected traits in cattle, play important roles in cattle breeding. Therefore, we carried out a genome-wide association study on birth weight and coat color to identify loci or potential linkage regions in 233 Simmental × Holstein crossbred beef cattle. The results revealed that nine SNPs were significantly associated with coat color (*rs137169378*, *rs110022687*, *rs136002689*, *Hypotrichosis_PMel17*, *PMEL_1*, *rs134930689*, *rs383170073*, *rs109924971*, and *rs109146332*), and these were in *RNF41*, *ZC3H10*, *ERBB3*, *PMEL*, and *OR10A7* on BTA5. Interestingly, *rs137169378*, *rs110022687*, *rs136002689*, *Hypotrichosis_PMel17*, and *PMEL_1* showed strong linkage disequilibrium (*r*^2^ > 0.8) and were significantly associated with coat color. Notably, *Hypotrichosis_PMel17* and *PMEL_1* were located in the gene *PMEL* (*p* = 2.22 × 10^−18^). Among the five significant SNPs associated with coat color, the birth weight of heterozygous individuals (AB) was greater than that of homozygous individuals (AA). Notably, the birth weight of heterozygous individuals with *Hypotrichosis_PMel17* and *PMEL_1* genotypes was significantly greater than that of homozygous individuals (0.01 < *p* < 0.05). Interestingly, the two loci were homozygous in black/white individuals and heterozygous in gray/white individuals, and the birth weight of heterozygous brown/white individuals (43.82 ± 5.25 kg) was greater than that of homozygous individuals (42.58 ± 3.09 kg). The birth weight of calves with the parental color (41.95 ± 3.53 kg) was significantly lower than that of calves with a non-parental color (43.54 ± 4.78 kg) (*p* < 0.05), and the birth weight of gray/white individuals (49.40 ± 7.11 kg) was the highest. Overall, *PMEL* appears to be a candidate gene affecting coat color in cattle, and coat color may have a selective effect on birth weight. This study provides a foundation for the breeding of beef cattle through GWAS for coat color and birth weight.

## 1. Introduction

Coat color is a distinguishing characteristic of domestic animal breeds, as it helps to distinguish them from other animals [1]. Coat color depends on melanin deposition, a trait that is expressed by fully differentiated melanocytes that secrete mature melanosomes. These, in turn, produce melanin and deliver it to keratinocytes to provide skin pigmentation, coat color, and feather color [2]. Melanosomes are discrete membranous organelles located within melanocytes; they are categorized into four stages based on function [3]. Stages I and II are unable to produce melanin and are known as premelanosomes. The premelanosomes are fibrillated and elliptical. At the beginning of phase III, melanin synthesis begins with the entry of tyrosinases (*TYP*, *TYRP1*) and ion-transporting proteins (*ATP7A*, *TPC2*) [4,5,6]. The premelanosomal proteins encoded by the premelanosome protein (*PMEL*) gene are responsible for the synthesis of fibers within the melanosome, and the fibers determine the shape of the melanosome [7,8] and are essential for melanin production. Therefore, *PMEL* has a crucial role in the formation of coat color.

In addition to being an important marker for distinguishing livestock breeds, coat color is involved in the regulation of physiological and growth traits. Coat color has been shown to directly affect the activation of thermoregulatory mechanisms [9]. Animals raised in cold weather usually have thicker coats to reduce heat loss than those in warm regions, and white animals seem to adapt better than black animals [10]. Isola [11] reported that the rectal and body surface temperatures of red and white Holstein cows were lower than those of black and white cows during the hot season. In addition, coat color has been associated with certain animal diseases, such as equine multiple congenital ocular anomalies (MCOA) syndrome in silver-coated horses [12]. Coat color is related to the production characteristics of domestic animals. Black-coated Holstein cows produce less milk than white-coated Holstein cows [13]. Cows with white coat color produced 394 kg more milk in 305 days compared to cows with predominantly black coats [14]. However, some studies have also concluded that the milk yield of Holstein cows was unrelated to or not influenced by coat color. In Menzies sheep, the growth traits of black wool sheep were consistently better than those of white wool sheep [15]. Birth weight in cattle is a trait with moderate heritability that responds to the development of the calf’s early embryo and later growth and development. As an important quantitative trait [16], birth weight is influenced by several factors. Most of the current studies on the factors influencing the birth weight of cattle have focused on breed [17], length of gestation, litter size [18], and season of birth [19]. The effect of coat color on birth weight has rarely been investigated.

Genome-wide association analysis (GWAS) offers the opportunity to identify genomic regions that are associated with phenotypic variation in economically important traits. A marker association of crossbred beef cattle using the highly parallel BovineSNP50 BeadChip (50K) assay identified five chromosomal regions that were associated with variation affecting birth weight and weight gain [20]. Three chromosomal regions and their genes affecting carcass quality traits in beef cattle have also been identified [21]. Rajawat et al. performed a genome-wide scan of 187 individuals from seven local cattle breeds and found that genes such as melanocortin 5 receptor (*MC5R*), myosin IA (*MYO1A*), and *PMEL* were associated with melanin synthesis, the biology of melanocytes and melanosomes, and migration and survival of melanocytes during development [22]. However, the relationship between coat color and birth weight in beef cattle has not been examined. Therefore, in this study, a whole-genome scan was performed on the F1 generation of Simmental × Holstein cattle to identify single-nucleotide polymorphisms (SNPs) or genomic regions associated with coat color and birth weight. The study provides basic information for exploring the relationship between coat color and bovine birth weight.

## 2. Materials and Methods

### 2.1. Animal Phenotyping and Blood Collection

Data were collected for birth weight and coat color traits from 238 German Simmental × Holstein cattle of the F1 generation from three ranches in Tianjin, China. For birth weight traits, basic statistics were recorded, and individuals with phenotypic missing values and those with phenotypic values outside plus or minus three times the standard deviation were excluded, while the coat color phenotype was classified by typical coat color and pattern.

The animal study protocol was performed in accordance with the regulations of the Chinese National Research Council (1994) and was approved by the Animal Care and Use Committee of the Institute of Animal Science, Chinese Academy of Agricultural Sciences (protocol code: IAS2019-83).

### 2.2. Genotyping and Quality Control

The genomic DNA of 233 individuals was extracted using a TIANamp Blood DNA Kit (Tiangen Biotech Company Limited, Beijing, China), and samples were genotyped using GGP Bovine 100K BeadChip. SNPs were analyzed using Genome Studio 2.0.

Quality control procedures were carried out by PLINK v1.9 [23,24] software as follows. First, we removed animals with more than 10% missing genotypes or SNP Mendel error rates > 2%. Then, we removed SNPs with call rates less than 95%, minor allele frequency (MAF) < 5%, <10% genotype appearance, or a Hardy–Weinberg equilibrium test *p* value < 10^−6^. Only autosomal SNPs with a known genome position according to the UMD_3.1 bovine assembly map were used. Missing genotypes were imputed using Beagle v3.3.2 [25,26]. Finally, a total of 233 individuals and 83,723 SNPs remained for GWAS.

### 2.3. Statistical Analyses

Considering the population stratification, PLINK was used for cluster analysis, and the first two principal components were selected as covariates in GWAS to correct population stratification and eliminate the influence of group stratification on the analysis results [27]. Here, a single-SNP GWAS was performed on the studied traits in crossbred data using the GEMMA v0.98.5 [28] software based on the following linear mixed model:(1)y=Xβ+Sα+Zμ+e
where *y* is a vector of phenotypic observations. *X* is the design matrix that relates the fixed effects to the observation, and β is a vector of fixed effects, including linear covariates of sex, herd, year of birth, and population stratification. *S* is the design matrix related to α, where α is a single SNP effect. *Z* is a design matrix relating observations to random animal genetic effects; μ is a vector of random additive polygenic effects, and *e* represents the residual error. The Wald F statistic was used to test the significance of effects for each SNP. For the GWAS of coat color, we divided the phenotypes into two groups (black/white = 1, red/white = 1, brown/white = 2, gray/white = 2 for the different crossbred calves’ coat colors; see Figure 1). The groups coded 1 and 2 represented individuals with the same and different coat colors as their parents, respectively.

### 2.4. Multiple Testing Corrections

The Bonferroni correction for multiple tests was used to determine *p* values. An SNP was deemed to be significant if the *p* value for the SNP was less than 0.05/N, where N is the total number of SNPs. The threshold value was 0.05/83,723 = 5.97 × 10^−7^ for coat color. However, the threshold of the genome-wide significance level for birth weight after the Bonferroni correction (0.05/83,723 = 5.97 × 10^−7^) was too strict, resulting in low statistical power. Therefore, a suggestive significance threshold for a *p* value of 1/83,723 = 1.19 × 10^−5^ was used in the current study for birth weight. The quantile–quantile (Q–Q) plots of the *p* value for each SNP were used to compare observed distributions of −log (*p* value) to the expected distribution under the null hypothesis for each trait. Manhattan plots of *p* values for each SNP were also used to illustrate significant associations at the level of each chromosome and trait. All plots were completed using the R v64 3.5.1 package qqman [29].

### 2.5. Mapping of Candidate Genes

The SNPs identified from the GWAS were mapped on the ARS-UCD1.2 genome assembly in the Ensembl Genome Browser database (https://asia.ensembl.org/, accessed on 15 September 2021). To avoid missing potentially important genes, we extended the confidence intervals by 100 kb on each side.

## 3. Results

### 3.1. Statistics and Analysis of Phenotypic Data

To examine the correlation between birth weight and coat color, 163 parental colors (black/white and red/white) and 70 non-parental colors (brown/white and gray/white) were plotted as probability density curves (Figure 1). The basic statistics of phenotype are listed in Table 1. The results showed that the birth weights of calves having the parental color (41.95 ± 3.53 kg) were significantly lower than those of calves with the non-parental color (43.54 ± 4.78 kg) (*p* < 0.05), and the birth weight of gray/white (49.40 ± 7.11 kg) individuals was the highest. To better show the different coat colors of the F1 generation, the offspring were photographed on-site, as shown in Figure 2.

### 3.2. Population Stratification Analysis of Offspring

To assess differences in the genetic backgrounds of hybrid individuals, a principal component analysis was performed on the F1 population (Figure 3). The first two principal components (PCs) contributed more than 85% of the variance and were, therefore, considered as covariates and included in the linear mixed model.

### 3.3. GWAS of Coat Color and Birth Weight

To identify the associated loci for birth weight and coat color, a GWAS of 233 F1 individuals was performed based on a mixed linear model, and Manhattan and Q–Q plots were obtained for the two traits (Figure 4a,b). For birth weight, four significant SNPs (*rs43326288*, *rs110100681*, *rs42712296*, and *rs109722507*) (*p* > [−log(*p*) = 4.92]) were screened, and these were distributed on BTA2, 10, 11, and 19, respectively. Among the SNPs, *rs42712296* and *rs109722507* were located in *ZC3H6* and *MSI2*, respectively. For coat color, a total of nine significant SNPs (*rs137169378*, *rs110022687*, *rs136002689*, *Hypotrichosis_PMel17*, *PMEL_1*, *rs134930689*, *rs383170073*, *rs109924971*, and *rs109146332*) were screened (*p* > [−log(*p*) = 6.22]), and all were distributed on BTA5. Six of these loci were located in the *RNF41*, *ZC3H10*, *ERBB3*, *PMEL*, and *OR10A7* genes (Table 2). Functional annotation of candidate genes was carried out to understand the effects of the genes on coat color and birth weight. The results showed that *MSI2* (promoting the proliferation and differentiation of muscle cells) was a candidate gene related to birth weight, and *ERBB3* and *PMEL* were candidate genes related to coat color. *PMEL* affected the deposition of melanin and was crucial to the formation of coat color (Table 2).

### 3.4. Association Analysis of Significantly Correlated SNPs for Coat Color and Birth Weight

GWAS analysis showed that there were nine SNPs on BTA5 that were significantly correlated with coat color. Linkage disequilibrium (LD) analysis of significant SNPs on BTA5 revealed that *rs137169378*, *rs110022687*, *rs136002689*, *Hypotrichosis_PMel17*, and *PMEL_1* exhibited high linkage imbalance (*r*^2^ > 0.8) (Figure 5a) and were mapped to a 232 kb region between positions 57,113,040 and 57,345,305. Among these, *hypotrichosis_PMEL17* and *PMEL_1* (*p* = 2.22 × 10^−18^) were located in *PMEL*.

To further understand the association between these five SNPs significantly related to coat color and birth weight, birth weight was measured and analyzed for significant SNPs in F1 populations of different genotypes (AA/BB = pure heterozygotes, AB = heterozygotes; the BB type only had one individual and, thus, was omitted). The results showed that there was no significant difference in birth weight between homozygous and heterozygous genotypes at *rs137169378*, *rs110022687*, or *rs136002689*, but the birth weight of heterozygous genotypes was higher than that of homozygous genotypes. The birth weights of heterozygous genotypes *Hypothesis_PMel17* and *PMEL_1* were significantly higher than those of homozygous genotypes (0.01 < *p* < 0.05) (Figure 5b).

To further understand the effects of *PMEL* on coat color and birth weight, genotypes at two significant SNP loci obtained from the GWAS analysis were statistically analyzed. The results showed that all black/white individuals were homozygous, and only one heterozygous genotype was found in red/white individuals. There were 32 homozygous genotypes and 34 heterozygous genotypes in brown/white individuals. There was no significant difference in birth weight between the two, but the birth weight of heterozygous individuals (43.82 ± 5.25 kg) was higher than that of homozygous individuals (42.58 ± 3.09 kg), and all gray/white individuals were heterozygous (Table 3). Combined with the previous results, the *PMEL* gene mutation apparently leads to a lighter coat color in beef cattle, and there is a certain selective effect on birth weight.

## 4. Discussion

Population stratification is a known confounding factor in GWAS, as it can lead to false positive results [37]. PCA analyses are widely used in population-level studies [38]. The PCA analysis of the F1 population from three pastures showed that there was stratification among the populations. Thus, it was necessary to employ a linear mixed model to improve the accuracy of the data and reduce false positives.

A GWAS for birth weight revealed that *rs42712296* and *rs109722507* were located in *ZC3H6* and *MSI2*, respectively. A study shows that *ZC3H6* is a negative regulator of macrophage activation and that it plays a role in cellular immune response [30]. *MSI2*-mediated *MiR7a-1* can promote differentiation of muscle stem cells and muscle regeneration in mice [31]. In terms of coat color, the gene annotation of the SNPs screened for significance revealed that these loci were located in the *RNF41*, *ZC3H10*, *ERBB3*, *PMEL*, and *OR10A7* genes. *ERBB3* encodes epidermal growth factor receptors, which can inhibit the development and differentiation of melanocytes [34]. Gutierrez-Gil et al. [39] analyzed the coat color of the backcross F2 population of Charolais and Holstein cattle and found that *ERBB3* and *PMEL* were candidate genes directly related to the pigmentation pathway. In addition, Pausch et al. [40] identified the Flavihe (German Simmental cattle) *ERBB3* locus as a potential candidate for ocular pigmentation in cattle and also found a significant correlation between *PMEL* and Simmental × Holstein coat color. *PMEL* is a pigment cell-specific protein, and animals lacking *PMEL* expression or expressing *PMEL* mutants exhibit varying degrees of hyperpigmentation or hypopigmentation [41]. Maibam et al. [42] found that the expression of the coat color genes *MC1R* and *PMEL* varied in different seasons, and *PMEL* was significantly correlated with coat color in Flavih cattle [43]. Other studies have shown that *PMEL* is related to the dilution of cow coat color [44,45]. In beef cattle, Satoshi Kimura [46] and others found that *PMEL p.Leu18del* caused brown Kumamoto cattle coat color dilution, and they also showed that this locus could be used as a DNA marker for cattle coat color variation in the *PMEL* of other species such as chickens [47], dogs [48], horses [49], and mice [50], where the SNP also caused the dilution of coat color. The current study shows that *PMEL* is also critical for coat color in cattle.

Coat color and birth weight are two of the most important traits that affect breed selection and adult weight in cattle. Birth weight is the most important trait of calves, as it is positively correlated with later growth traits and negatively correlated with cow dystocia [51,52]. Although birth weight, as a quantitative trait, is affected by many factors, SNPs have been found to affect birth weight in recent years [53]. Research has shown that the mutation of the *IGFBP-3* intron region had a significant effect on the birth weight of hybrid cattle, and the birth weight and body weight of heterozygous animals were higher than those of homozygous animals [54]. As an important trait, coat color has a certain selective effect on the production traits of cattle. It has been reported that the weight of cattle with a deep coat color is significantly higher than that of cattle with a light coat color in high-temperature environments [55]. In white pigs, the birth weight of piglets with *KIT* dominant genotype *II* was significantly higher than that of piglets with the recessive *i* allele.

The coding region of a gene encodes a protein that determines the function of the gene. If the coding region is mutated (especially by a missense mutation), changes in protein structure or function may occur, thereby affecting gene function. Single amino acid mutations can lead to protein aggregation, misfolding, and dysfunction [56]. In this study, SNPs significantly associated with coat color were identified, among which *Hypothesis_PMel17* and *PMEL_1* were located in the *PMEL* gene. Both SNPs were located in the CDS region of the *PMEL* gene as missense mutations that altered the amino acid sequence, resulting in individuals with gray/white coat color, among others, in the F1 population. Statistical analysis of genotypes at *Hypothesis_PMel17* and *PMEL_1* showed that black/white individuals were homozygous, gray/white individuals were heterozygous, and the birth weight of heterozygous brown/white individuals was higher than that of homozygous individuals. According to the statistics of five SNPs (highly linked *r*^2^ > 0.8), genotype was significantly related to coat color; the birth weight of heterozygous individuals was higher than that of homozygous individuals, especially for the two significant loci within *PMEL.* The birth weight of heterozygous calves was significantly higher than that of homozygous calves. At the same time, the statistics for coat color and birth weight showed that the birth weight of non-parent-colored individuals was significantly higher than that of parent-colored individuals, and the birth weight of gray/white individuals was the highest. These results indicate that mutation of *PMEL* has a dilution effect on coat color, and coat color may have a selective effect on the birth weight of calves. This result provides new ideas for subsequent breeding work.

## 5. Conclusions

In this study, coat color and birth weight analyzed by GWAS obtained nine and four significant SNPs, respectively. The *PMEL* of *Hypothesis_PMel17* and *PMEL_1* loci are on BTA5. Black/white individuals were homozygous, gray/white individuals were heterozygous, and the birth weight of heterozygous (brown/white individuals 43.82 ± 5.25 kg) was higher than that of homozygous individuals (42.58 ± 3.09 kg). At the same time, the birth weight of heterozygous individuals for five highly linked SNPs was higher than that of homozygous individuals, especially the birth weight of calves heterozygous for *Hypothesis_PMel17* and *PMEL_1*, which was significantly higher than that of the corresponding homozygous genotype. The birth weight of calves with the parental color (41.95 ± 3.53 kg) was significantly lower than that of calves with a non-parental color (43.54 ± 4.78 kg), and the birth weight of gray/white (49.40 ± 7.11 kg) individuals was the highest. Our findings suggested that *PMEL* may be a candidate gene affecting coat color in cattle and that coat color may have a selective effect on birth weight.

## Figures and Tables

**Figure 1 animals-13-03821-f001:**
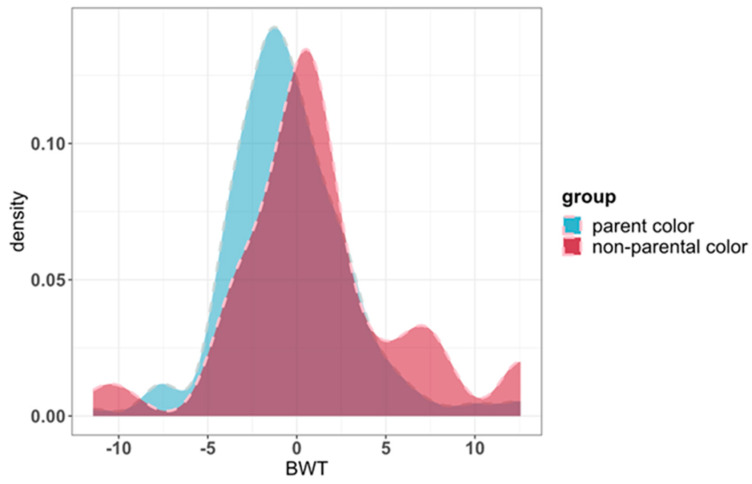
Probability distributions of birth weight according to coat color.

**Figure 2 animals-13-03821-f002:**
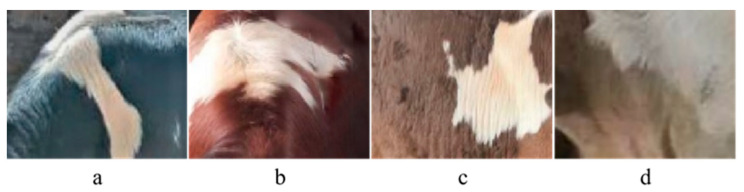
Coat color categories used for categorizing the F1 population in Simmental × Holstein calves. (**a**) black/white. (**b**) red/white. (**c**) brown/white. (**d**) gray/white.

**Figure 3 animals-13-03821-f003:**
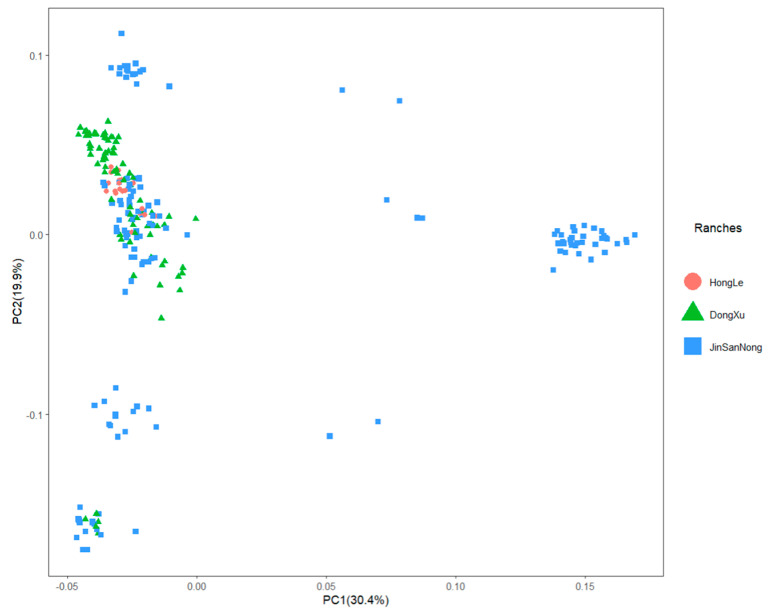
PC plot for crossbred cattle.

**Figure 4 animals-13-03821-f004:**
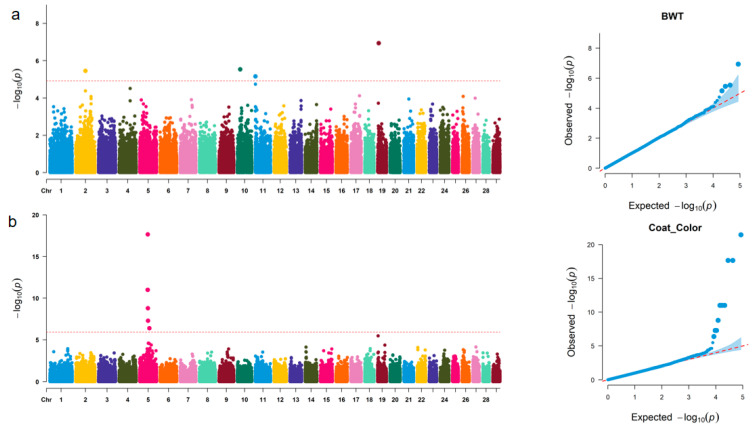
Locations of significant SNPs and annotation results. (**a**) Manhattan and quantile–quantile (Q–Q) plots of GWAS for birth weight in crossbred cattle. (**b**) Manhattan and quantile–quantile (Q–Q) plots of GWAS for coat color in crossbred cattle.

**Figure 5 animals-13-03821-f005:**
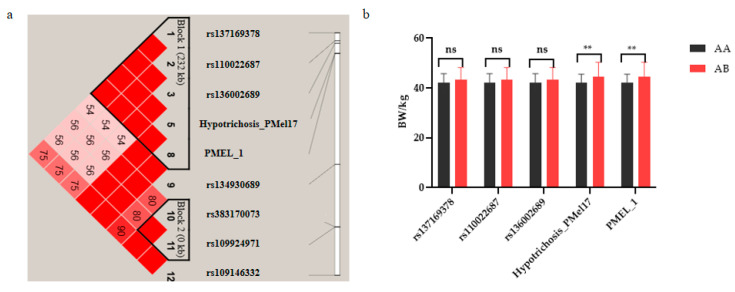
Analysis of SNPs significantly associated with coat color and birth weight. (**a**) Linkage disequilibrium analysis of candidate regions associated with coat color. (**b**) Birth weights of individuals with different genotypes at significant SNPs associated with coat color in crossbred cattle. Note: The number of individuals with genotype AA of *rs137169378*, *rs110022687*, and *rs136002689* was 174, and the number with genotype AB was 58. The number of individuals with genotype AA of *Hypotrichosis_PMEL17* and *PMEL*_*1* was 196, and the number of individuals with genotype AB was 37. ** Represents 0.01 < *p* < 0.05.

**Table 1 animals-13-03821-t001:** Number and average birth weight per category of coat color in crossbred cattle.

Coat Color	Number	Birth Weight (kg)	
black/white	158	41.92 ± 3.48	41.95 ± 3.53 ^a^
red/white	5	42.80 ± 4.53	
brown/white	66	43.18 ± 4.32	43.54 ± 4.78 ^b^
gray/white	4	49.40 ± 7.11	
Total/Mean	233	42.47 ± 3.95	

Note: Different letters represent significant differences (*p* < 0.05).

**Table 2 animals-13-03821-t002:** The significant SNPs associated with birth weight and coat color of crossbred cattle and their annotated candidate genes and gene functions.

Traits	Chromosome	SNP Reference	Significant SNPs	Position	*p* Value	Candidate Genes	Gene Names	Login Number	Gene Function
Birth weight	2	*rs43326288*	BovineHD0200019955	68801335	3.49 × 10^−6^	*-*	*-*	*-*	*-*
	10	*rs110100681*	ARS-BFGL-NGS-89500	20284816	2.91 × 10^−6^	*-*	*-*	*-*	*-*
	11	*rs42712296*	BovineHD1100000027	188097	6.93 × 10^−6^	*ZC3H6*	Zinc finger CCCH-type containing 6	XM_0598916151	Activation of macrophages and cellular immune response [30].
	19	*rs109722507*	BovineHD1900002375	8344354	1.15 × 10^−7^	*MSI2*	Musashi RNA binding protein 2	NM_001206656.2	Promote the proliferation and differentiation of muscle cells [31].
Coat color	5	*rs137169378*	BovineHD0500016234	57113040	1.01 × 10^−11^	*RNF41*	Ring finger protein 41	NM_001046525.1	Dampen inflammatory responses and promote tissue repair [32].
	5	*rs110022687*	BovineHD0500016256	57201800	1.01 × 10^−11^	*ZC3H10*	Zinc finger CCCH-type containing 10	NM_001097993.1	Regulation of mitochondrial function and promotion of myoblast differentiation [33].
	5	*rs136002689*	BovineHD0500016261	57230765	1.01 × 10^−11^	*ERBB3*	Erb-b2 receptor tyrosine kinase 3	NM_001103105.1	Inhibits melanocyte maturation [34].
	5	*-*	Hypotrichosis_PMel17	57345303	2.22 × 10^−18^	*PMEL*	Premelanosome protein	XM_005206504.5	Affects melanosome maturation and melanin deposition [35].
	5	*-*	PMEL_1	57345305	2.22 × 10^−18^				
	5	*rs134930689*	BovineHD0500016585	58599028	7.01 × 10^−13^	*-*	*-*	*-*	*-*
	5	*rs383170073*	5-59657291-A-C-rs383170073	59305325	3.13 × 10^−10^	*OR10A7*	Olfactory receptor family 10 subfamily A member 7	NM_001389764.1	Responsible for recognition and G protein-mediated odor signaling [36].
	5	*rs109924971*	BovineHD0500016696	59306323	3.13 × 10^−10^	*-*	*-*	*-*	*-*
	5	*rs109146332*	BovineHD0500016831	59851727	1.18 × 10^−18^	*-*	*-*	*-*	*-*

**Table 3 animals-13-03821-t003:** The birth weights of calves with two significant SNPs located in *PMEL* under different coat colors.

Coat Color	SNP	Genotypes	Number	Birth Weight (kg)
black/white	*Hypotrichosis_PMel17*	GG	158	41.92 ± 3.48
		CG	0	0
	*PMEL_1*	AA	158	41.92 ± 3.48
		CA	0	0
red/white	*Hypotrichosis_PMel17*	GG	4	42.50 ± 5.25
		CG	1	44.00 ± 0.00
	*PMEL_1*	AA	4	42.50 ± 5.25
		CA	1	44.00 ± 0.00
brown/white	*Hypotrichosis_PMel17*	GG	32	42.58 ± 3.09 ^a^
		CG	34	43.82 ± 5.25 ^a^
	*PMEL_1*	AA	32	42.58 ± 3.09 ^a^
		CA	34	43.82 ± 5.25 ^a^
gray/white	*Hypotrichosis_PMel17*	GG	0	0
		CG	4	49.40 ± 7.11
	*PMEL_1*	AA	0	0
		CA	4	49.40 ± 7.11

Note: The same letter represents no difference (*p* > 0.05).

## Data Availability

The data presented in this study are available on request from the corresponding author. The data are not publicly available due to data from this study need to be followed up with further analysis.

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
