# Peer review of "Genome-Wide Association Analysis Identifies the *PMEL* Gene Affecting Coat Color and Birth Weight in Simmental × Holstein"

_animals, 2023, doi:10.3390/ani13243821_

Round 1

Reviewer 1 Report

Comments and Suggestions for Authors

General Comments

This research delves into the genetic intricacies of coat color and birth weight in Simmental×Holstein cattle through a comprehensive genome-wide association study. The study identifies specific SNPs associated with coat color, particularly those linked to the PMEL gene, shedding light on the genetic factors influencing these key traits. This research holds promise for selective breeding programs, offering genetic markers that could optimize breeding strategies for desired coat colors and potential impacts on birth weight. Beyond its practical applications, the study significantly contributes to the scientific knowledge of cattle genetics, providing valuable insights for both researchers and stakeholders in the livestock industry. However, there are certain areas that should be addressed to enhance the manuscript's potential for publication.

Specific comments:

Point 1: The authors detailed several candidate genes associated with the investigated phenotypes. What rationale led to the exclusive focus on the PMEL gene in the title?

Point 2: The manuscript uses the notation SNP>A>C>G. This notation might be unclear or confusing for readers. Are there specific reasons for using this notation? If not, consider removing it for clarity.

Point 3: Did the authors conduct population stratification analysis using PCA or MDS? This step is important in GWAS analysis and should be addressed.

Point 4: When referring to cattle populations like Holstein, Menzies, Simmental, etc., the authors consistently use italic font. Could the authors provide a rationale for employing this formatting choice?

Point 5: Please include the effect numbers for fixed effects in the statistical analysis.

Point 6: While stating the number of SNPs in the manuscript, such as 83723, the authors consistently use spaces within the number throughout the entire manuscript. Please either remove the spaces or use commas for better consistency.

Point 7: When preparing the manuscript, please ensure to use the appropriate format when explaining the p-value.

Point 8: Please ensure that the software versions mentioned in the manuscript are formatted correctly, for example, Beagle v3.3.2, PLINK v1.9, R v3.5.1, etc., and provide appropriate references for each.

Point 9: Include the phenotypic distribution plot, and in Table 1, present the basic phenotypic statistics related to birth weight.

Point 10: The images presented in Figure 1 lack clarity. Please provide the figure with a higher resolution for better visibility. Additionally, the title of Figure 1 appears confusing, and its formatting is inconsistent with that of Figure 2. Ensure that the titles are appropriately formatted and accurately reflect the content of each figure.

Point 11: The presentation of results in sections 3.1, 3.2, and 3.3 is unclear and confusing for readers. Please revise and provide clearer explanations to enhance the understanding of the results.

Point 12: Please provide a new table containing detailed information on the identified significant genes.

Point 13: Did the authors conduct functional annotations for the identified significant candidate genes in the GWAS study? Please provide detailed explanations.

Point 14: The conclusions section in this manuscript requires improvement to provide a clearer explanation of the study's results, its significance, and the implications for future research.

Author Response

Revision list according to the comments from Reviewer

General Comments:

This research delves into the genetic intricacies of coat color and birth weight in Simmental×Holstein cattle through a comprehensive genome-wide association study. The study identifies specific SNPs associated with coat color, particularly those linked to the PMEL gene, shedding light on the genetic factors influencing these key traits. This research holds promise for selective breeding programs, offering genetic markers that could optimize breeding strategies for desired coat colors and potential impacts on birth weight. Beyond its practical applications, the study significantly contributes to the scientific knowledge of cattle genetics, providing valuable insights for both researchers and stakeholders in the livestock industry. However, there are certain areas that should be addressed to enhance the manuscript's potential for publication.

Reply:

Thank you very much for your affirmation of this study. As you said, this study explored the potential impact of coat color on birth weight through GWAS, and provided new ideas for animal husbandry researchers with rich knowledge of bovine genetics. Thank you again for your evaluation. Your comments are the greatest encouragement to me and also give good guidance for my future research. At the same time, thank you for your question, which makes my research more perfect. I will seriously consider your questions and carefully revise them.

Specific comments:

Point 1: The authors detailed several candidate genes associated with the investigated phenotypes. What rationale led to the exclusive focus on the PMEL gene in the title?

Reply:

Thank you very much for your suggestions. The aim of this paper was to investigate the relationship between coat color and birth weight. GWAS results showed that SNPs significantly associated with coat color were localised to several genes such as PMEL, and genotypically heterozygous individuals at the significant loci all had higher birth weight than genotypically pure individuals. In contrast, PMEL encodes pre-melanosomal proteins that are responsible for the synthesis of fibres within the melanosome and determine the shape of the melanosome, which is essential for melanin production. Several studies have also shown that PMEL has an important effect on coat color (see Introduction and Discussion). PMEL is therefore highlighted.

Point 2: The manuscript uses the notation SNP>A>C>G. This notation might be unclear or confusing for readers. Are there specific reasons for using this notation? If not, consider removing it for clarity.

Reply:

Thank you very much for your suggestions. Changes have been made to the rs numbers in the article, see lines 26-26, 188-192, 245 and 254. Already marked in yellow in the article.

Point 3: Did the authors conduct population stratification analysis using PCA or MDS? This step is important in GWAS analysis and should be addressed.

Reply:

Thank you very much for your suggestions. We have made changes in the article to add PCA analysis. Already marked in yellow in the article.

Point 4: When referring to cattle populations like Holstein, Menzies, Simmental, etc., the authors consistently use italic font. Could the authors provide a rationale for employing this formatting choice? 

Reply:

Thank you very much for your suggestions. We have corrected the italics of the individual species names. Already marked in yellow in the article.

Point 5: Please include the effect numbers for fixed effects in the statistical analysis.

Reply:

Thank you very much for your suggestion. I'm sorry I didn't understand the effect numbers for fixed effects. Does it mean the number of fixed effects? In this study, four fixed effects were taken into account in the linear model: farms sex PC1 PC2. In a linear model of birth weight, the beta values of the four fixed effects are -0.277056, -2.26098, -4.84625, and -7.79646.

Point 6: While stating the number of SNPs in the manuscript, such as 83723, the authors consistently use spaces within the number throughout the entire manuscript. Please either remove the spaces or use commas for better consistency.

Reply:

Thank you very much for your suggestion. We have standardised the formatting of the punctuation in the text to indicate quantities. See lines 142-145, 247. Already marked in yellow in the article. 

Point 7: When preparing the manuscript, please ensure to use the appropriate format when explaining the p-value.

Reply:

Thank you very much for your suggestion. The p-value format has been standardised. Already marked in yellow in the article. 

Point 8: Please ensure that the software versions mentioned in the manuscript are formatted correctly, for example, Beagle v3.3.2, PLINK v1.9, R v3.5.1, etc., and provide appropriate references for each.

Reply:

Thank you very much for your suggestion. The software name has been written correctly. Literature [29] is also cited for additional information. See lines 113, 119, 150. Already marked in yellow in the article.

Point 9: Include the phenotypic distribution plot, and in Table 1, present the basic phenotypic statistics related to birth weight.

Reply:

Thank you very much for your suggestion. Phenotype distribution maps have been added, see Figure 1. We counted the birth weight of crossbred cattle, and did not count the basic phenotypic data related to birth weight.

Point 10: The images presented in Figure 1 lack clarity. Please provide the figure with a higher resolution for better visibility. Additionally, the title of Figure 1 appears confusing, and its formatting is inconsistent with that of Figure 2. Ensure that the titles are appropriately formatted and accurately reflect the content of each figure.

Reply:

Thank you very much for your suggestion. The clarity of Figure 1 has been adjusted and the formatting has been harmonised with Figure 2. Already marked in yellow in the article.

Point 11: The presentation of results in sections 3.1, 3.2, and 3.3 is unclear and confusing for readers. Please revise and provide clearer explanations to enhance the understanding of the results.

Reply:

Thank you very much for your suggestion. 3.1, 3.2 and 3.3 have been rewritten. Already marked in yellow in the article.

Point 12: Please provide a new table containing detailed information on the identified significant genes.

Reply:

Thank you very much for your suggestion. To re-add table 2.

Point 13: Did the authors conduct functional annotations for the identified significant candidate genes in the GWAS study? Please provide detailed explanations.

Reply:

Thank you very much for your suggestion. To re-add table 2.

Point 14: The conclusions section in this manuscript requires improvement to provide a clearer explanation of the study's results, its significance, and the implications for future research.

Reply:

Thank you very much for your suggestion. The conclusion section has been written. Already marked in yellow in the article.

Reviewer 2 Report

Comments and Suggestions for Authors

In this study, the authors identified potential loci and mutations/SNPs which were significantly associated with the coat color and birth weight traits in SimmentalxHolstein cattle using GWAS technology. After analysis, they found that two missense mutations (rs57345303G>C and rs57345305A>C) within PMEL gene were notably associated with coat color, and individuals with heterozygous genotype have higher birth weight than those with homozygous genotypes, which was considered as candidate gene for coat color and birth weight in cattle. It is an interesting research but several major concerns still exist, leading to that it cannot be accepted in the present form.

Major concerns:

Overstatement, such as line 37-38, 90-91, 206, and all the description involved in gray/white pattern cattle.

In line 97-98: Statistics generally believe that the phenotypic values over or below mean ± 2-fold SD should be excluded, while here the author set it as mean ± 3-fold SD, why? Is there any reference basis.

In Table 1, whether the birth weight traits among four coat color cattle are statistically significant, and the significant symbols should be added. In addition, the individual numbers of red/white and gray/white cattle are too less to accurately evaluate the association between genotype and phenotype. Similarity, the individual numbers in Fig3b should also be presented, and only based on four individuals is not enough to association analysis and indicate the phenotype of heterozygous cattle (Table 2).

How does the coat color influence birth weight in cattle, on maternal or fetal? Additionally, why did the heterozygous individuals (rather than one of the homozygous ones) have lighter coat color and birth weight traits? At least, they should be discussed in the Discussion section.  

Comments on the Quality of English Language

Moderate editing of English language required.

Author Response

Revision list according to the comments from Reviewer 

General Comments:

In this study, the authors identified potential loci and mutations/SNPs which were significantly associated with the coat color and birth weight traits in SimmentalxHolstein cattle using GWAS technology. After analysis, they found that two missense mutations (rs57345303G>C and rs57345305A>C) within PMEL gene were notably associated with coat color, and individuals with heterozygous genotype have higher birth weight than those with homozygous genotypes, which was considered as candidate gene for coat color and birth weight in cattle. It is an interesting research but several major concerns still exist, leading to that it cannot be accepted in the present form.

Reply:

Thank you very much for your affirmation of this study. As you said, this study explores the potential impact of hair color on birth weight through GWAS, which is a very interesting study. Thank you again for your evaluation. Your comments are the greatest encouragement to me and also give good guidance for my future research. At the same time, thank you for your question, which makes my research more perfect. I will seriously consider your questions and carefully revise them.

Overstatement, such as line 37-38, 90-91, 206, and all the description involved in gray/white pattern cattle.

Reply:

Thank you very much for your suggestion. Adjustments have been made to lines 37-38, 90-91, 206 in the original text; see lines 40-41, 93-94, 309-310. Gray/white individuals were also redescribed. Already marked in yellow in the article.

In line 97-98: Statistics generally believe that the phenotypic values over or below mean ± 2-fold SD should be excluded, while here the author set it as mean ± 3-fold SD, why? Is there any reference basis.

Reply:

Thank you very much for your suggestion. According to the normal distribution, about 95% of the values are distributed within a range of 2-fold SD from the mean, and about 99.7% of the values are distributed within a range of 3-fold standard deviations from the mean. In order to better and more clearly demonstrate the authenticity of individual data, we chose to obtain as many individuals as possible, so we chose mean±3-fold SD.

In Table 1, whether the birth weight traits among four coat color cattle are statistically significant, and the significant symbols should be added. In addition, the individual numbers of red/white and gray/white cattle are too less to accurately evaluate the association between genotype and phenotype. Similarity, the individual numbers in Fig3b should also be presented, and only based on four individuals is not enough to association analysis and indicate the phenotype of heterozygous cattle (Table 2).

Reply:

Thank you very much for your suggestion.The significance symbol has been added in Table 1, and the quantity indicated in the original Figure 3b (now figure 5b) is shown in the note. We have noticed that there are too few red white and gray white individuals, so we have compared the non parent color (brown/white and gray/white) with the parent color (black/white and red/white), as shown in table 1. At the same time, the original table 3 is deleted and table 3 is added to further explain the role of PMEL.

How does the coat color influence birth weight in cattle, on maternal or fetal? Additionally, why did the heterozygous individuals (rather than one of the homozygous ones) have lighter coat color and birth weight traits? At least, they should be discussed in the Discussion section.

Reply:

Thank you very much for your suggestion. We add to this part of the discussion, see lines 276-308. Already marked in yellow in the article.

Moderate editing of English language required.

Reply: 

Thank you very much for your suggestion. We sent the article to a professional English touch-up agency (LetPub) and revised the entire document for sentence structure and language.

Round 2

Reviewer 1 Report

Comments and Suggestions for Authors

Dear Authors, I have reviewed your manuscript and commend the revisions made according to the comments and suggestions provided during the first revision. I appreciate your thorough attention to detail in addressing the comments, which significantly improved the quality and clarity of the manuscript. Based on the revisions, I am pleased to accept the article for publication. Thank you for your efforts.

Reviewer 2 Report

Comments and Suggestions for Authors

None.